# Evaluating pictorial support in person-centred care for children (PicPecc): a protocol for a crossover design study

Stefan Nilsson [1], Angelica Wiljén [2], Jonas Bergquist [3], John Chaplin [4], Ensa Johnson [5], Katarina Karlsson [6], Tomas Lindroth [7], Anneli Schwarz [2], Margaretha Stenmarker [4,8,9], Gunilla Thunberg [10], Linda Esplana,[11] Eva Frid,[12] Malin Haglind,[11] Angelica Höök,[13] Joakim Wille,[12] Joakim Öhlen [1,14]

► Prepublication history and supplemental material for this paper is available online. To view these files, please visit the journal online (http://dx.doi.org/10.1136/bmjopen-2020-042726).

For numbered affiliations see end of article.

**Correspondence to**
Dr Stefan Nilsson;
stefan.nilsson.4@gu.se

## ABSTRACT

**Introduction** This study protocol outlines the evaluation of the pictorial support in person-centred care for children (PicPecc). PicPecc is a digital tool used by children aged 5–17 years to self-report symptoms of acute lymphoblastic leukaemia, who undergo high-dose methotrexate treatments. The design of the digital platform follows the principles of universal design using pictorial support to provide accessibility for all children regardless of communication or language challenges and thus facilitating international comparison.

**Methods and analysis** Both effect and process evaluations will be conducted. A crossover design will be used to measure the effect/outcome, and a mixed-methods design will be used to measure the process/implementation. The primary outcome in the effect evaluation will be self-reported distress. Secondary outcomes will be stress levels monitored via neuropeptides, neurosteroids and peripheral steroids indicated in plasma blood samples; frequency of in-app estimation of high levels of distress by the children; children's use of analgesic medicine and person centeredness evaluated via the questionnaire Visual CARE Measure. For the process evaluation, qualitative interviews will be carried out with children with cancer, their legal guardians and case-related healthcare professionals. These interviews will address experiences with PicPecc in terms of feasibility and frequency of use from the child's perspective and value to the caseworker. Interview transcripts will be analysed using an interpretive description methodology.

**Ethics and dissemination** Ethical approval was obtained from the Swedish Ethical Review Authority (reference 2019-02392; 2020-02601; 2020-06226). Children, legal guardians, healthcare professionals, policymaking and research stakeholders will be involved in all stages of the research process according to Medical Research Council's guidelines. Research findings will be presented at international cancer and paediatric conferences and published in scientific journals.

**Trial registration** ClinicalTrials.gov; NCT04433650.

### Strengths and limitations of this study

► A person-centred framework is used for the design of the intervention.
► A child-centred pictorially supported communication device is used for self-report from the age of 5 years.
► The study evaluates a complex intervention with a combination of self-reported symptoms and biomarkers.
► Biomarkers of stress are monitored via blood plasma.
► The process evaluation will give additional information for future usage based on the frequency and feasibility of use.

## INTRODUCTION

Children with cancer struggle with several physical and emotional symptoms. Their ability to communicate these symptoms is dependent on various factors such as age, maturity, diagnosis, cognitive status, psychological status, language ability and cultural background as well as situational aspects. Alleviating distress caused by cancer is beneficial for both children, their families and healthcare professionals[1]. Symptom identification and communicative support can enable symptom relief with the potential to reduce distress and alleviate suffering for the child and will also improve quality of the care.[1]

### Person-centred care for children

Person-centred care is found in ethics and based on the assumption that every person has resources that should be used in the care situation; being human is about having capabilities. This can be referred to Homo capax,[2] that is, a person with capabilities and vulnerabilities, and who is considered responsible for his/her actions in relationships with others.[3]

There is no gold standard definition of person-centred care and the exploration of the concept has emphasised many different aspects and different definitions. In this project, the definition of person-centred paediatric care is based on three key concepts of partnership, narrative and documentation, generating a cocreated partnership and safeguarding the partnership through documenting the child's narrative, preferences and participation.[4 5]

The project is found on the ethical principles put forward by the French philosopher Paul Ricœur, which aims for the good life, with and for others, with equitable and unbiased institutions.[6] In this regards, a person-centred approach with a child perspective not only includes the idea of what is best for the child but also acknowledges the self-determination of the child. Decisions are, therefore, made that balance these concepts; that is, neither solely from an adult's view of the child's needs nor solely from the perspective of the children themselves. Instead, the desired solution is to combine the child's experience, the perspectives of legal guardians and significant others and the healthcare professionals. Within this balance, however, it is important to always prioritise the children's best interests in an attempt to optimise their well-being.[7]

To initiate a person-centred approach for paediatric care is to elucidate, listen to and affirm the child's narrative. Assessments of symptoms are essential in symptom relief for children with cancer, and self-reports are the gold standard for measuring symptoms.[8 9]

Children's own voices and self-reports are necessary to our understanding of the issues facing children if we are to reach the goal of symptom relief. [7] Children with cancer—like all children—have the right to actively take part in decisions regarding their health. In order to achieve this, they need support to communicate issues related to their symptoms. For such a system to work well in their everyday lives, symptom communication will largely rely on identification of symptoms, and communication skills and pathways to present this information in a timely and appropriate manner within their healthcare management.[10]

## Universal design

Healthcare professionals often tend to use language that is too complex for children to understand. Children can, therefore, be said to be 'communication vulnerable',[11] depending on their level of health literacy, potential cognitive or communicative disabilities, age, language level or competency in the majority language.[12] The Convention on the Rights of Persons with Disabilities put forth the idea of 'universal design' to the design of products, environments, programmes and services, so that they would be usable for all people, to the greatest extent possible, without the need for adaptation or specialised knowledge.[13]

The application of new digital technologies using pictorial supported communication may assist communication vulnerable children in healthcare to more effectively self-report and communicate with others about their symptoms, overcoming their and possibly their families' communication difficulties. Pictorial communication support may foster closer relationships, trust and more open communication between families and healthcare professionals.[14]

## Self-assessment tools

The development of assessment tools for children to self-report pain started in the 1980s, with the widespread implementation of these tools in the 1990s.[15] However, children's self-reports have been shown to still fail to impact healthcare, and there is a need for innovative ideas that support the implementation of these assessment tools in clinical practice.[16 17] Enabling children with cancer to self-report their symptoms may help them to understand their condition better and thereby better cope with their illness. Communicating symptoms in an effective way that can quickly alert healthcare professionals to their discomfort are empowering process that will make them feel secure in knowing that they have strategies that give them the possibility to communicate with somebody who will assist them to achieve symptom relief.[18]

Although validated patient self-report instruments exist for some symptoms, healthcare professionals seldom use these in clinical practice;[17] furthermore, most paediatric conditions lack a validated symptom assessment tool. What is missing from the clinical toolbox is an instrument that assesses the intensity of symptoms in a simple, valid and reliable way.[19] One of the few symptoms that is assessed in clinical practice is pain intensity. Smeland *et al* found that, overall, pain was assessed using a validated tool in 19% of the children in post anaesthesia care; this fell to 9% in children aged <5 years old.[17] An explanation for this could be either that these instruments do not exist or that they are difficult to use, interpret or unreliable. Healthcare professionals prefer to rely on personal judgement and experience with the patient and family[20] and, therefore, the measurement process must contribute to this and not try to replace it. The use of an instrument that focuses on a single symptom, for example, pain intensity, does not adequately capture the overall experience of the child and can be considered a restrictive application. Novel assessment tools that give a broader description of symptoms are, therefore, needed in order that the child can fully communicate their experience.

## Distress in children

The term 'distress' refers to a multifactorial unpleasant emotional experience that can be described as a combination of fear, anxiety and pain.[21] The relationships between these factors are complex, and the experience of distress is based on interactions between 'genetically linked behaviour patterns, temperamental predispositions, normal developmental fears, parental psychopathology and discrete learning experiences'.[22] Distress in this study is defined as an experiential response and sensation of the mind associated with negative emotions that appear when

a situation is fearful or impossible to manage from the perspective of the child. Distress can be a consequence of insufficient symptom relief, and self-reported distress is a global assessment that reflects the child's experience of the success of symptom relief. It is important to evaluate the distress in children undergoing cancer treatment and to find strategies for the measurements of symptoms/emotions that are reliable and valid for this purpose. It is known that acute stress activates the hypothalamic–pituitary–adrenal axis and the sympathetic nervous system as well as the hypothalamic–pituitary–gonadal axis.[23–25] For example, plasma cortisol concentration is an established stress (energy mobilisation) indicator that is known to react within minutes after the onset of stress exposure. Estradiol is on the other hand an anabolic hormone, which protects against adverse effects of stress.

### The medical scenario within which PicPecc will be tested
The drugs used to treat children with cancer can lead to several negative side effects, for example, children undergoing cancer treatment frequently report nausea and vomiting and other kinds of distress.[26] One of the drugs that is used in cancer treatment is methotrexate, which is one of the most effective medications in the treatment of acute lymphoblastic leukaemia (ALL) in children.[27] High-dose methotrexate is used world wide and has been included as part of the Nordic Society for Paediatric Haematology and Oncology ALL treatment protocols since 1981.[28] Furthermore, the treatment is given according to a strictly detailed Nordic and European schedule, that is, clinical conditions have been well established. For these reasons, treatment with high-dose methotrexate has been chosen as the medical context within which the effect of the use of person-centred care for children (PicPecc) tool will be evaluated from a person-centred perspective.

The primary aim of the project is to investigate whether a person-centred communication intervention through the use of the PicPecc digital communication tool for children undergoing cancer treatment decreases the children's symptom-related distress in general. A secondary aim is to investigate the process of implementing person-centred communication through the use of PicPecc tool.

### Main research question
Does adding the PicPecc tool decrease distress (measured on an 11-point numeric rating scale (NRS) (0 (no distress) and 10 (worst possible distress)) in children with ALL, aged 5–17 years, who undergo high-dose methotrexate treatment?

### Secondary research questions
1. Does the application of the PicPecc tool, increase person-centredness measured on Visual CARE Measure (VCM) in children with ALL, aged 5–17 years, who undergo high-dose methotrexate treatment?
2. Does the application of the PicPecc tool alter stakeholders' perspectives in a positive direction towards person-centred communication?

### Hypothesis
1. Children undergoing cancer treatment will experience lower distress levels, when they can report their holistic symptoms in a system created using universal design principles (ie, the PicPecc tool with pictorial support) than will children with standard healthcare communication opportunities (the primary outcome). In addition to a decrease in self-reported stress levels, there will also be a decrease in neuropeptides, neurosteroids and peripheral steroids for stress and pain.
2. Person-centred care is enhanced, through enabling children to proactively assess their symptoms from a holistic perspective and communicate these to their healthcare providers within an enhanced communication framework (ie, using the PicPecc tool).

## METHODS AND ANALYSIS
### Study design
The Medical Research Council's key principles and actions for development and evaluation of complex interventions[29 30] guided the intervention development and the research design. In a hybrid design, both the effects of the intervention and the implementation process will be evaluated.[31] Relevant care situations are selected[32] where highly standardised care procedures are used and where there is a range of different situations where children struggle with symptoms. To facilitate the effect evaluation, the children will participate in a crossover design study where they are their own controls (figure 1). The study design follows the SPENT 2019 checklist for clinical trials.[33]

The development of the PicPecc tool follows established guidelines[34] and was based on the theoretical framework of person-centred care,[4] on published systematic reviews[9] and on systematic reviews conducted within the project on assessment tools for nausea,[35] and anxiety.[36] Children with cancer, their legal guardians and healthcare professionals have been involved throughout the development process. The study protocol outlined here pertains to the evaluation and implementation phases.

### Participants and units
#### Context and setting
In Sweden, approximately 350 children are diagnosed with cancer each year. The treatment of childhood cancer is conducted at six childhood cancer centres and at regional hospitals.[37] Three of these childhood cancer centres and five regional hospitals in Sweden will participate in the study.

### Selection criteria
Inclusion criteria are children diagnosed with ALL, between 5 and 17 years of age whose treatment plan includes at least two treatments of high-dose methotrexate. The child needs to have a cognitive level of at least 5 years (ie, to be able to understand an NRS).[38] The

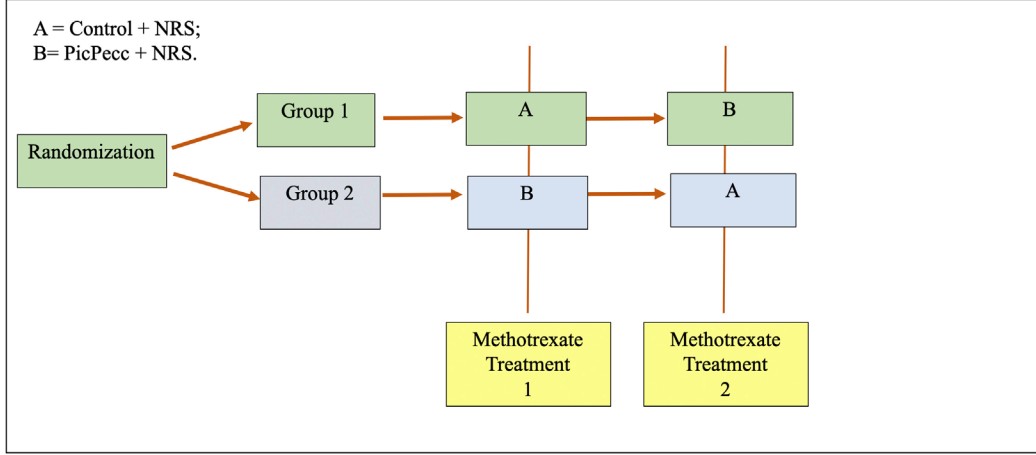

**Figure 1** The cross-over design with two study groups will participate in two phases as related to the Nordic and European study protocols in Sweden for treatment of children with high-dose methotrexate. All methotrexate treatment sessions take a similar amount of time for the child. The intervals between each of the methotrexate treatments will be controlled by each child's treatment plan and may vary between 3 and 6 weeks. NRS, numeric rating scale; PicPecc, pictorial support in person-centred care for children.

child's understanding of an NRS will be tested before inclusion based on a situational judgement test which involves a realistic, hypothetical scenario about a child who fell from a tree. The child will be asked to assess pain using the NRS. This situational judgement test has previously been validated to discriminate positive and negative emotions.[39] Exclusion criteria are children 0–4 years, no verbal assent from children unable to read, scheduled to undergo only one high-dose methotrexate treatment.

The inclusion criteria for legal guardians will be that their child has undergone the high-dose methotrexate treatment and has used the PicPecc tool. In addition, legal guardians will need to be at the hospital during the treatment.

The inclusion criteria for healthcare providers will be that they are responsible for the children's care during the high-dose methotrexate treatment when the children use the PicPecc tool.

### Method of recruitment

The recruitment is planned to start at the beginning of 2021. The surveyed children participate in data collection two times, once as a control (A) and once at the time of symptom reporting and initial use of the communication tool PicPecc (B) (figure 1). Children with cancer aged 5–17 years old, legal guardians and healthcare professionals at three childhood cancer centres and five regional hospitals in Southern Sweden will participate in the study. Each year approximately 175–200 children get cancer in Southern Sweden, and about a third of these children receive a diagnosis of ALL. At the three childhood cancer centres and at the five regional hospitals included in this study, approximately 25 of these children will fulfil the inclusion criteria for this study each year.

Healthcare providers at each of the units will be interviewed. The nurse and/or nurses who initiate and conclude the high-dose methotrexate treatment will be invited to a semistructured interview.

### Consent process

Legal guardians of children below 15 years of age with ALL who are scheduled to receive high-dose methotrexate treatments will be informed about the study by a physician or a nurse, included in the research group. The legal guardian will receive written information, and the child will be given text and picture-based information. On consent from a legal guardian, child assent is obtained verbally and in writing if the child can read. Older children (aged 15 years or above) will give written consent themselves. In Sweden, children between 15 and 18 years can provide written informed consent themselves if they are assessed to have the level of maturity and capacity to understand the consequences of participation.

### Randomisation

The participants will be allocated codes in a consecutive order; the code is randomly assigned to either the intervention phase (B) or control phase (A). Participating cancer units will be given the solution to the randomisation code once the codes have been allocated to the participants. The participants will only have access to the PicPecc tool during the intervention phase. There will be a period of at least 2 weeks between the end of the first intervention period and the start of the next methotrexate treatment. This provides an adequate washout period between the intervention (B) and the control phase (A) for any behaviour change to revert to previous patterns. Following the transtheoretical model,[40] habitual behaviour change related to health is a process involving a number of stages that takes time to complete successfully, where support for change is removed early in the process the individual will quickly revert to previous habituated behavioural patterns.

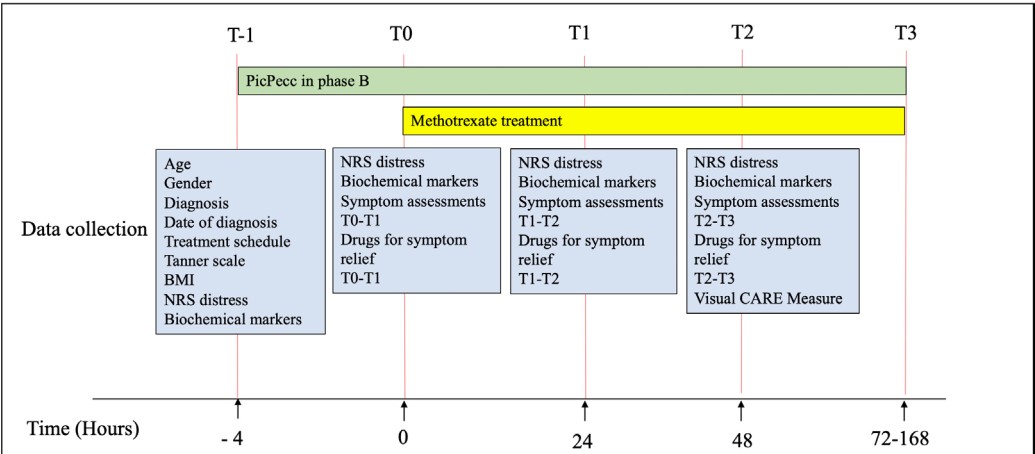

Flow chart Methotrexate treatment (A=Baseline and B=Intervention)

**Figure 2** Data collection time points and variables in both the control and intervention phases. BMI, body mass index; NRS, numeric rating scale; PicPecc, pictorial support in person-centred care for children.

We are confident, therefore, that the intervention will have no residual effects on the control phase.

## Measures and materials

### Impact evaluation

We consider a difference of 15% to be a meaningful difference in score average between T0 and T2 (48 hours); this is represented by a difference of approximately 1.5 units on the NRS (0–10) of distress, when comparing users of the PicPecc tool to control subjects. The estimate of SD is based on unpublished data of 11 to 12-year-old girls' self-reports.[41] Based on an expected SD of 2.9 (and a power of 0.8), it is necessary to include at least 32 participants. With a dropout rate of approximately 20%, 20 participants in each group, that is, 40 participants will be included in the study.

In both the control (A) and intervention phases (B), the data collection follows the test period outline in figure 2. Assessment of distress will be made at time points T-1, T0, T1 and T2. T3 is an interview to evaluate the implementation process. Primary outcome is the difference in delta T0 and T2 between control and intervention phases. The time points are linked to the schedule for the methotrexate treatment to avoid extra blood sampling. The time points will also facilitate the evaluation between before and after treatment, with the objective to evaluate differences in symptoms, with and without the PicPecc tool (figure 2).

Primary outcome:

The change in the primary outcome variable (distress) from baseline (T0) to 48 hours after treatment start (T2) measured on an 11-point NRS (0 (no distress) and 10 (worst possible distress))[42 43] will be compared between the control and intervention phases. Self-reported distress (NRS-11) will also be collected 4 hours before high-dose methotrexate (T-1) and after 24 hours (T1) (figure 2) in order to establish within subject variation.

Secondary outcomes:

1. Blood samples will be collected and steroid levels in plasma will be monitored. Pain and steroid levels in blood: neuropeptides, neurosteroids and peripheral steroids

will be collected before start (T-1 and T0) of the high-dose methotrexate treatment, 24 hours after start (T1) and 48 hours after start (T2). Since blood-drawing procedures are part of routine monitoring of cancer care, a small sample of the blood will be obtained for this research, with no additional needle pricks required. Steroids are measured in this study using Liquid chromatography tandem mass spectrometry and Super critical fluid chromatography tandem mass spectrometry (LC-MS/MS and SFC-MS/MS) methods.[44] It is not possible to distinguish between different types of stress, but the design includes the evaluation of two indicators of stress response, first, biological (measured by biomarkers), and second, perceived (self-reported). Since the same individual is assessed before and after the chemotherapy, both with and without the PicPecc tool, it is possible to evaluate the effect of the PicPecc tool on intervention-related stress.

2. Self-reported person centredness. This is evaluated on the VCM,[45] which will be collected 48 hours after the start (T2) of the high-dose methotrexate treatment. The VCM provides the legal guardians of children <7 years old (VCM 10Q-legal guardians), children aged 7–11 years (VCM 5Q) and adolescents aged 12 years and over (VCM 10Q) the opportunity to report their experiences regarding both the meeting with the healthcare professional and their participation in decision related to healthcare.[45]

3. Frequency of assessments of symptoms with the PicPecc tool. In-app assessment levels will be recorded during the intervention phase, and during the control phase, a checklist will be used (eg, frequency of symptom assessments T0–T2 (figure 2)).

Drug consumption for all types of symptom relief. These data will be collected from the patients' medical records.

### Process evaluation

After each intervention, experiences of care during the treatment are explored in individual semistructured

interviews (T3) with all participating children, their legal guardians and the healthcare professionals involved in the children's care. The objective is to illuminate the experiences of using the PicPecc tool from the perspective of the participating children, their legal guardians and the healthcare professionals. The interviews will be thematically analysed following the procedures of Braun and Clarke[46] to give an understanding of how the PicPecc tool was used during the intervention.

Numerical data regarding when and how often the children used the PicPecc tool will also be collected.

The semistructured interviews will follow an interview guide adapted for the child according to age and maturity. The questions will also be provided with pictorial support according to the concept of universal design (online supplemental file 1). The child and the legal guardian are interviewed separately. The aim is to interview both the child and their legal guardian; however, if either of them is unwilling or does not fit the criteria the other (child or legal guardian) will be invited to participate on their own.

### Intervention

In the intervention phase, the child will use the PicPecc tool before and during high-dose methotrexate treatment for communicative support to assess their symptoms and emotions. The PicPecc tool is used in a phone or a tablet computer; delivered via an iOS or Android platform. The development of the PicPecc tool is presented elsewhere.[34] The PicPecc tool is based on a child-centred assessment approach, and the goal is to adapt the assessment to the child's age, maturity, diagnosis, language ability and cultural background. All sections of the PicPecc tool will contain pictures, text and sound. The PicPecc tool includes an assessment scale, which is designed as a thermometer. The thermometer is graded from 0 (green) to 10 (red). Each level of the scale is also symbolised with a face that shows the intensity of each symptom and/or emotion. The result of the assessment is visualised as a facial expression and colour that represents the intensity of the symptom and/or emotion (ie, anxiety, appetite, fear, how I am feeling today, nausea, pain and sleep). In addition, the PicPecc tool has a body outline without any markings on which the child can indicate the location of the symptom and/or emotion, pictures for the type of symptom and/or emotion as well as open questions where the child can write narratives about symptoms and/or emotions. The child receives visual feedback from the app and directly from healthcare professionals participating in the intervention, on their reported assessments made with the thermometer. The child can follow the assessments on an hourly, daily or weekly basis. The PicPecc tool also includes a personal avatar to represent the child. Using the avatar the child can make choices of the avatar's gender, skin and hair colour, and its facial expressions thus contributing to the inclusiveness of the PicPecc tool by providing racial and gender diversity. The avatar will

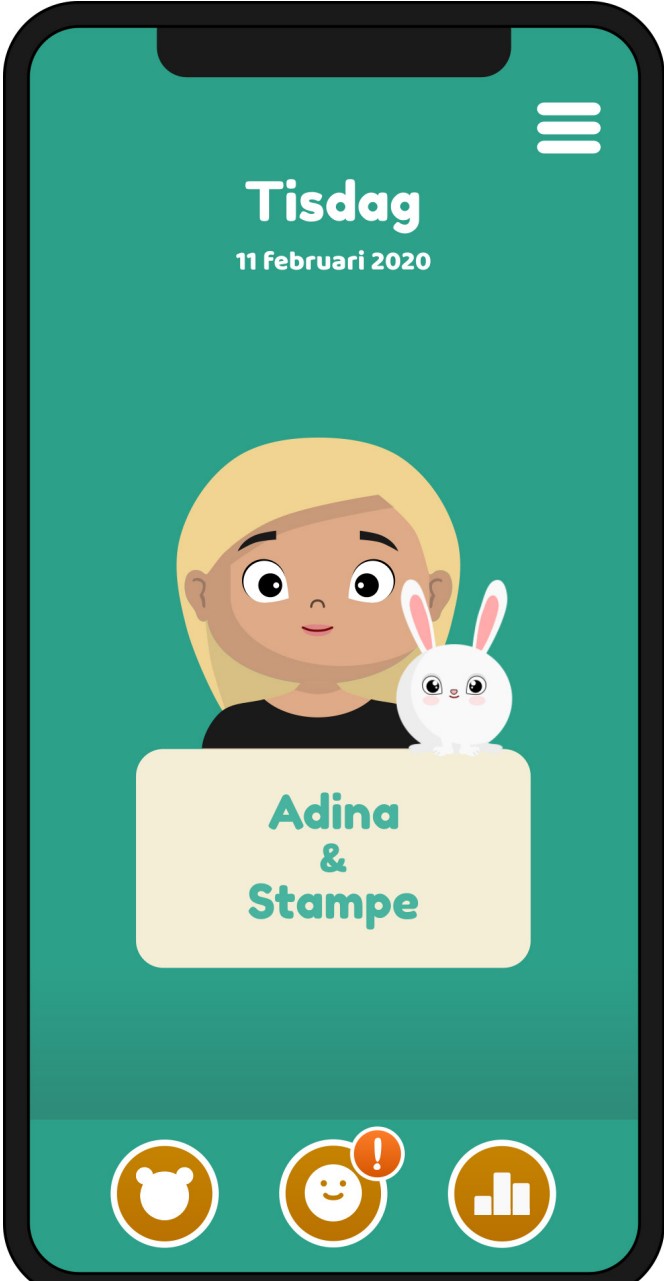

**Figure 3** The PicPecc tool consists of an avatar and pets that the child can win through interaction with the reporting process. PicPecc, pictorial support in person-centred care for children.

be linked to the child throughout all the assessments. In addition, in order to enhance interaction with the tool the design of the app includes a gamification element, for example, the child will get a reward in the form of a pet when he/she has assessed the symptoms and/or emotions (figures 3–5). The child is encouraged with a reminder in the PicPecc tool to assess the symptoms and/or emotions two times per day (in the morning and in the evening). The child can in addition assess his/her symptoms and/or emotions more frequently with the PicPecc tool, for example, if he or she prefers to do so.

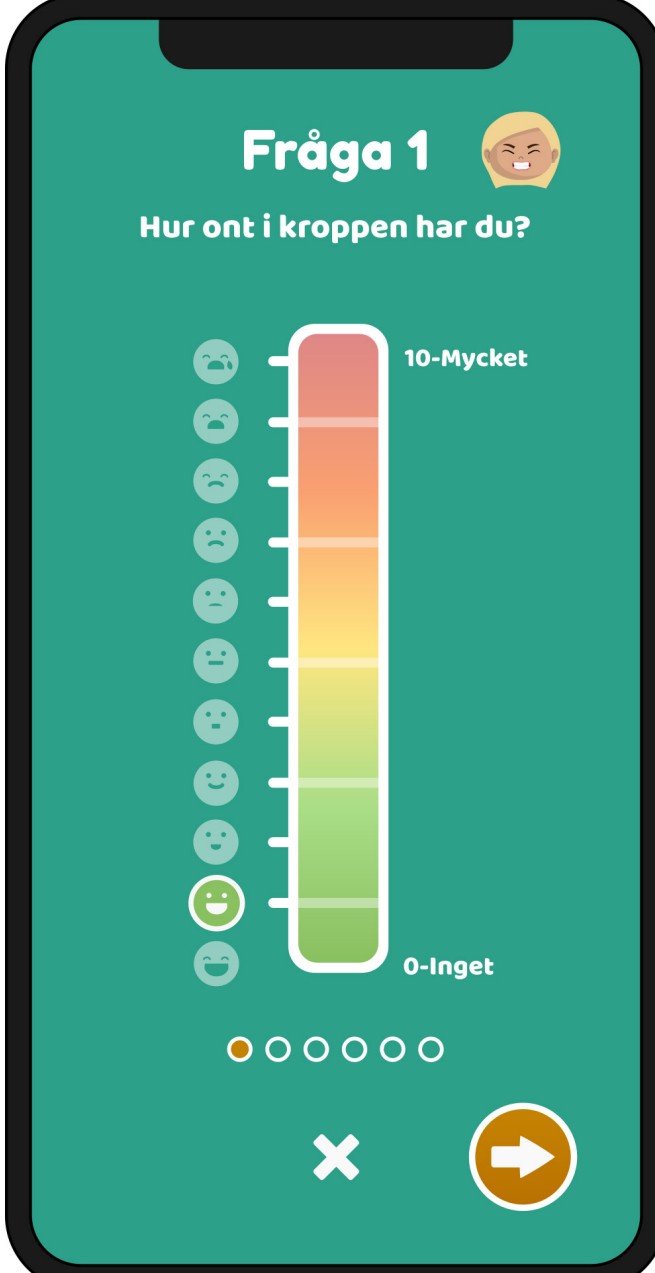

**Figure 4** Reports are made by using a thermometer for assessing symptoms and emotions.

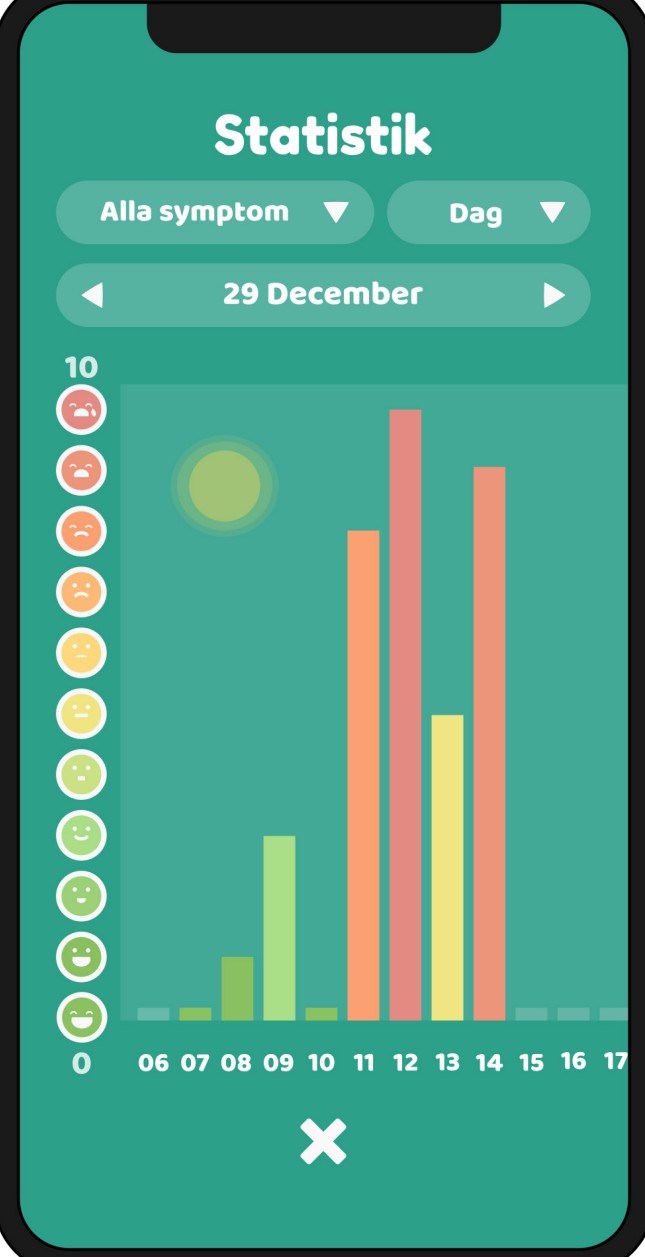

**Figure 5** The child receives feedback on the assessments in the form of diagrams showing the results of the latest days or weeks.

### The implementation of a person-centred approach in both phase A and phase B

The implementation strategies of person-centred communication consist of two components:

1. A person-centred workshop for the paediatric oncology teams about enhanced symptom communication.
2. One member of the research team will be assigned to coach their colleagues in each of the clinical departments on the person-centred approach. They will support the implementation of the intervention and be responsible for data collection.
3. Workshops with paediatric oncology teams.

Paediatric oncology teams will be invited to a workshop that will scrutinise and discuss communication issues based on five questions (figure 6). These questions will be: how communication, based on a person-centred approach, can be implemented in clinical practice in child healthcare?

### Clinical coaches to support the implementation of the intervention

One coach in each of the clinical departments will support the implementation of the intervention. The coach will be responsible for facilitating education and support for their colleagues in the clinical department. In addition, the coaches will also be responsible for data collection.

| Communication | | |
|---|---|---|
| 1 | What is communication based on a person-centred approach? | |
| | 2 | How can universal design facilitate communication? |
| **Symptom management** | | |
| | 3 | What is distress (in blood and as self-reports)? |
| | | 4 What is symptom management and treatment goals? |
| | | 5 What is the PicPecc tool? |

**Figure 6** The questions will be discussed step by step in the workshops with communication and symptom management. PicPecc, pictorial support in person-centred care for children.

The coach in each clinical ward will get support with the research process from the research group.

## Procedural fidelity

The procedural fidelity will be evaluated in each phase. The coach will (a) monitor the occurrence of relevant variables, (b) provide documentation that the experimental conditions occurred as planned, (c) provide support to practitioners about the use of the interventions.

## Data analyses plan

### Effect evaluation

We expect the intervention to be superior to the control in terms of the health outcome assessment (NRS-11). We also expect that there will be a difference between premethotrexate treatment (T1, T0) and methotrexate treatment (T1, T2) in both intervention and control phases. Therefore, we will test the null hypothesis that there will be neither change in any of the measurements between the premethotrexate treatment (T-1, T0) and methotrexate treatment (T1, T2) nor between intervention and control phases. A p value of $<0.05$ will be considered as statistically significant. Categorical data will be descriptively analysed by frequency distributions and percentages. The paired sample t-test will evaluate the difference between two sets of assessments and effect size.[47] Data will be analysed with IBM SPSS Statistics V.25 (New York City, USA).

## Process evaluation

The qualitative data analysis will be driven by interpretive description methodology, and the analysis will follow a mixed-methods research design, that is, a convergent design, with concurrent timing where qualitative and quantitative data are independent of each other. The goal is to disclose experiential and contextually shaped knowledge.[48] The qualitative data will be interpreted, and the analysis will lead to the identification of a set of themes, which describe the participant's experience of using the tool. The quantitative data about the frequencies of the participants' use of the PicPecc tool will be analysed with descriptive statistics, which will then be integrated with the qualitative analysis to facilitate a deeper understanding of how the participants use the PicPecc tool. Finally, an interpretation will be conducted between qualitative and quantitative data.[49]

## Patient and public involvement

Children with cancer, legal guardians and their healthcare professionals have been involved in the development of the PicPecc tool.[34] Healthcare professionals have been involved in the development of the hybrid design, in order to optimise the feasibility of the study.

## Data monitoring committee

The study will have an external expert panel that will be responsible for checking the quality of the data in the study. The expert panel will also evaluate ethical issues that emerge during the study period.

## Ethics and dissemination

### Ethics

Ethical approval was obtained from the Swedish Ethical Review Authority (ref 2019-02392; 2020-02601; 2020-06226) for the planned studies. Children are a vulnerable group since adults have a power relationship with the child, the child with cancer is in a difficult life situation, and the child is expected to share personal stories. All data collection is carried out during hospital treatment, and all ordinary management and safety mechanisms are in place. If complications occur in conjunction with the intervention, these are reported at the usual clinical rounds and will be managed according to the ordinary routines.

The children and their legal guardians will be informed about the purpose of the study. The information to participants states that all participation is voluntary and will not adversely affect the child's healthcare, and that it is possible to withdraw consent without explanation or any negative consequences on their treatment and care. All

data will be kept confidential, and it is only the research group that has access to the data. The results will not reveal the identity of the participants. Research with children, legal guardians and healthcare professionals requires an ethics committee approval, written consent from the child and the legal guardian and verbal assent from children unable to read.

## Dissemination

Research findings will be presented at international cancer and paediatric conferences, published in scientific journals and publications for children with cancer and their legal guardians. The results will also be available for professional training purposes.

### Author affiliations
[1]University of Gothenburg Centre for Person-Centred Care, and Institute of Health and Care Sciences, University of Gothenburg, Gothenburg, Sweden
[2]Department of Paediatrics, Södra Älvsborg Hospital, Region Västra Götaland, Borås, Sweden
[3]Department of Chemistry – Biomedical Centre, Analytical Chemistry and Neurochemistry, Uppsala University, Uppsala, Sweden
[4]Department of Paediatrics, Institute for Clinical Sciences, Sahlgrenska Academy, University of Gothenburg, Gothenburg, Sweden
[5]Centre for Augmentative and Alternative Communication, University of Pretoria, Pretoria, South Africa
[6]Faculty of Caring Science, Work Life and Social Welfare, University of Borås, Borås, Sweden
[7]Division of Informatics, University of Gothenburg, Gothenburg, Sweden
[8]Department of Paediatrics, Region Jönköping County, Jönköping, Sweden
[9]Department of Clinical and Experimental Medicine, Linköping University, Linköping, Sweden
[10]DART centre for Augmentative and Alternative Communication and Assistive Technology, Sahlgrenska University Hospital, Region Västra Götaland, Gothenburg, Sweden
[11]Queen Silvia Children's Hospital, Sahlgrenska University Hospital, Region Västra Götaland, Gothenburg, Sweden
[12]Department of Paediatric Oncology and Haematology, Skåne University Hospital, Lund, Sweden
[13]Department of Anaesthesia and Intensive Care, Linköping University Hospital, Linköping, Sweden
[14]Palliative Centre, Sahlgrenska University Hospital, Region Västra Götaland, Gothenburg, Sweden

**Contributors** Conceptualisation: SN, AW, JB, JC, TL, MS, AH, JÖ. Funding acquisition: SN, AW, JB, JC, EJ, KK, TL, AS, MS, GT, JÖ. Methodology: SN, JB, JC, MS, JÖ. Project administration: SN. Supervision: SN, JB, JC, MS, JÖ. Visualisation: SN. Writing the original draft: SN, AW, JB, JC, EJ, KK, TL, AS, MS, GT, LE, EF, MH, AH, JW, JÖ.

**Funding** This work was supported by Barncancerfonden, grant number TJ2017-0028, KP2018-0023, MTI2019-0011; Vinnova, grant number MTI2019-0011; STINT, Vetenskapsrådet, Forte, grant number SA2018-7681, South Africa-Sweden University Forum and the University of Gothenburg Centre for Person-centred Care (GPCC), Sweden, which is funded by the Swedish Government's grant for Strategic Research Areas (Care Sciences) and the University of Gothenburg, Sweden, no specific grant number.

**Competing interests** None declared.

**Patient consent for publication** Not required.

**Provenance and peer review** Not commissioned; externally peer reviewed.

includes any translated material, BMJ does not warrant the accuracy and reliability of the translations (including but not limited to local regulations, clinical guidelines, terminology, drug names and drug dosages), and is not responsible for any error and/or omissions arising from translation and adaptation or otherwise.

### ORCID iDs
Stefan Nilsson http://orcid.org/0000-0002-8847-9559
Angelica Wiljén http://orcid.org/0000-0002-7463-1628
Jonas Bergquist http://orcid.org/0000-0002-4597-041X
John Chaplin http://orcid.org/0000-0001-8128-4225
Ensa Johnson http://orcid.org/0000-0001-6203-1433
Katarina Karlsson http://orcid.org/0000-0003-4080-6677
Tomas Lindroth http://orcid.org/0000-0002-8404-2376
Anneli Schwarz http://orcid.org/0000-0003-1102-3948
Margaretha Stenmarker http://orcid.org/0000-0002-9631-5757
Gunilla Thunberg http://orcid.org/0000-0002-9582-7814
Joakim Öhlen http://orcid.org/0000-0003-2429-8705

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
