## [Reviewer comments · BMJ Open]

ARTICLE DETAILS

TITLE (PROVISIONAL)	Evaluating Pictorial support in Person-centred Care for Children (PicPecc): a protocol for a crossover design study
AUTHORS	Nilsson, Stefan; Wiljén, Angelica; Bergquist, Jonas; Chaplin, John; Johnson, Ensa; Karlsson, Katarina; Lindroth, Tomas; Schwarz, Anneli; Stenmarker, Margaretha; Thunberg, Gunilla; Esplana, Linda; Frid, Eva; Haglind, Malin; Höök, Angelica; Wille, Joakim; Öhlen, Joakim

VERSION 1 – REVIEW

REVIEWER	Murray, Janice Manchester Metropolitan University, Faculty of Health, Psychology and Social Care
REVIEW RETURNED	01-Nov-2020

GENERAL COMMENTS	Thank you for inviting me to review this paper. The proposed work is very welcome and the study protocol is clear. Importance has been given to universal design and visual communications systems. The team have drawn on an appropriate range of reference material. I could only spot one typographical error. page 10, lines 58/9 - either remove the word 'a' or the plural maker on the word 'tools'. I look forward to reading an article on the findings of the study in due course.
---

REVIEWER	Linder, Lauri University of Utah, College of Nursing
REVIEW RETURNED	23-Nov-2020

GENERAL COMMENTS	Overall, the protocol is clearly written and organized. I did have some comments for additional clarification and to strengthen the manuscript. Introduction (Page 2). The authors speak of the importance of prioritizing the children's best interests. Please provide additional clarity as to how this is accomplished or proposed strategies for ensuring this happens. Regarding the distress measures, particularly the physiologic measures of distress. How do you proposed to distinguish between emotional aspects of distress and additional physiologic distress related to the physical
--

	consequences of cancer and chemotherapy treatment? The primary aim speaks of a decrease in children’s distress symptoms. Please clarify as to whether you are speaking of symptom-related distress in general or distress scores for individual symptoms. Study sample – are you including just those receiving high dose methotrexate for acute lymphoblastic leukemia or might children with other diagnoses – e.g., osteosarcoma be eligible? Are you considering only children with a primary diagnosis of cancer – e.g., not with relapsed disease or secondary cancers? Please provide justification for including such a broad age range 5-17 years. Do you have preliminary data addressing the feasibility and acceptability of the resource across this wide range of ages and developmental abilities? Please provide clarity regarding the number of potentially eligible children each year out of the 350 who are diagnosed each year in Sweden. Regarding the consent/assent process – is 15 the age of majority in Sweden for providing informed consent? If so, you may wish to include this to add clarity for international readers, many of whom are more familiar with age 18 as the age at which adolescents are able to provide their own written informed consent independent of their parents. Study procedure – The protocol as presently stated is unclear as to how frequently participants will be asked to interact with the tool. What are the expectations for daily use, and how many interactions is each participant anticipated to have? How is the PicPecc tool delivered – e.g., phone or tablet. Is it delivered via an iOS or Android platform? Please add clarity regarding the basis for the measurement intervals in the study. Please provide clarity as to how differences will be detected in the qualitative analyses. Please also provide justification for the use of the cross-over design for the study. How confident can you be that the intervention effects will not carry over for those who were randomized to use the PicPecc tool first?
--	--

VERSION 1 – AUTHOR RESPONSE

Reviewer 1

Comments

I could only spot one typographical error. page 10, lines 58/9 - either remove the word 'a' or the plural maker on the word 'tools'.

Answers

We have revised the text to "... pain was assessed using a validated tool in 19% of the children..." Amendent on page 9, line 3.

Reviewer 2

Comments

Regarding the distress measures, particularly the physiologic measures of distress. How do you proposed to distinguish between emotional aspects of distress and additional physiologic distress related to the physical consequences of cancer and chemotherapy treatment?

Answers

It is not possible to distinguish between different types of stress, but the design includes the evaluation of two indicators of stress response, firstly, biological (measured by biomarkers) and secondly, perceived (self-reported). Since the same individual is assessed before and after the chemotherapy, both with and without the PicPecc tool, it is possible to evaluate the effect of the PicPecc tool on intervention-related stress. Amendent on page 16, lines 10-15.

Comments

The primary aim speaks of a decrease in children's distress symptoms. Please clarify as to whether you are speaking of symptom-related distress in general or distress scores for individual symptoms.

Answers

The primary aim of the project is to investigate whether a person-centred communication intervention through the use of the PicPecc digital communication tool for children undergoing cancer treatment decrease the children's symptom-related distress in general. Amendent on page 10, line 23.

Comments

Study sample – are you including just those receiving high dose methotrexate for acute lymphoblastic leukemia or might children with other diagnoses – e.g., osteosarcoma be eligible? Are you considering only children with a primary diagnosis of cancer – e.g., not with relapsed disease or secondary cancers?

Answers

Inclusion criteria are children diagnosed with acute lymphoblastic leukaemia, between 5 and 17 years of age whose treatment-plan includes at least two treatments of high-dose methotrexate, including children with relapsed disease. Amendent on page 13, lines 6-7.

Comments

Please provide clarity regarding the number of potentially eligible children each year out of the 350 who are diagnosed each year in Sweden.

Answers

Each year approximately 175-200 children get cancer in Southern Sweden, and about a third of these children receive a diagnosis of ALL. Amendent on page 13, lines 23-25.

Comments

Regarding the consent/assent process – is 15 the age of majority in Sweden for providing informed consent? If so, you may wish to include this to add clarity for

international readers, many of whom are more familiar with age 18 as the age at which adolescents are able to provide their own written informed consent independent of their parents.

Answers

In Sweden, children over 15 years must provide written informed consent in addition to that provided by their legal guardians if they have the capacity to understand the consequences of participation. Amendent on page 14, lines 8-10.

Comments

Study procedure – The protocol as presently stated is unclear as to how frequently participants will be asked to interact with the tool. What are the expectations for daily use, and how many interactions is each participant anticipated to have?

Answers

The child is encouraged with a reminder in the PicPecc tool to assess the symptoms and/or emotions twice daily (in the morning and in the evening). The child can in addition assess his/her symptoms and/or emotions more frequently with the PicPecc tool, e.g., if he or she prefers to do so. Amendent on page 18, lines 19-22.

Comments

How is the PicPecc tool delivered – e.g., phone or tablet. Is it delivered via an iOS or Android platform?

Answers

The PicPecc tool is used in a phone or a tablet computer; delivered via an iOS or Android platform. Amendent on page 17, line 22-23.

Comments

Please add clarity regarding the basis for the measurement intervals in the study.

Answers

The time points are linked to the schedule for the methotrexate treatment to avoid extra blood sampling. The time points will also facilitate the evaluation between before and after treatment, with the objective to evaluate differences in symptoms, with and without the PicPecc tool Amendent on page 15, lines 14-17.

Comments

Please provide clarity as to how differences will be detected in the qualitative analyses.

Answers

The objective is to illuminate the experiences of using the PicPecc tool from the perspective of the participating children, their legal guardians and the healthcare professionals. The interviews will be thematically analysed following the procedures of Braun and Clarke to give an understanding of how the PicPecc tool was used during the intervention. Amendent on page 17, lines 7-11.

Comments

Please also provide justification for the use of the cross-over design for the study. How confident can you be that the intervention effects will not carry over for those who were randomized to use the PicPecc tool first?

Answers

The participants will only have access to the PicPecc tool during the intervention phase. There will be a period of at least two weeks between the end of the first intervention

period and the start of the next methotrexate treatment. This provides an adequate washout period between the intervention (B) and the control phase (A) for any behaviour change to revert to previous patterns. Following the transtheoretical model habitual behaviour change related to health is a process involving a number of stages which takes time to complete successfully. Where support for change is removed early in the process the individual will quickly revert to previous habituated behavioural patterns. We are confident therefore, that the intervention will have no residual effects on the control phase. Amendent on page 14, lines 16-24.

VERSION 2 – REVIEW

REVIEWER	Linder, Lauri University of Utah, College of Nursing
REVIEW RETURNED	27-Dec-2020

GENERAL COMMENTS	Thank you for the added clarity around the inclusion of children with ALL during high dose methotrexate treatment. Thank you for the added clarity around symptom-related distress as the outcome of the primary aim of the study. Regarding the second secondary research question, when you speak of “alter stakeholders’ perspectives ...” are you proposing a specific direction in relation to these altered perspectives? Might you reframe this question slightly as to give attention to the desired direction? Please provide justification regarding the inclusion of children with relapsed disease. Please provide inclusion/exclusion criteria regarding the parents/guardians and healthcare providers. Will parents and children be enrolled as dyads? If a child is eligible yet a parent does not meet enrollment criteria, would the child be allowed to participate in the study. Which members of the pediatric oncology team are eligible to participate in the study and how will they be recruited? Please address the following clarifying questions in relation to the informed consent/assent process:  • To clarify, will the children 15-17 years of age provide written consent along with written permission from their parents? In Sweden, is this regarded as written consent or written assent? • Are the children less than 15 years of age required to provide any type of written assent or is verbal assent sufficient? Please note that some of these questions relate to typical processes in the US in which parents provide written permission for the child’s participation (for children 17 years and younger). Technically, many argue that parents truly cannot provide informed consent for the child so the preferred expression is permission. In many centers, children 7-17 years of age provide written assent in addition to their parents’ written permission for their participation. Once the individual is 18 years of age, he/she
--

	is able to provide his/her own informed consent independent of the parents. Thank you for providing information about the typical prevalence of children newly diagnosed with ALL in southern Sweden each year. Of the children with ALL, approximately how many are anticipated to be eligible to participate in the study, and what is the anticipated recruitment/enrollment rate? Please provide clarity regarding the intended frequency for use of the PicPecc tool during the intervention period. Is there an intended frequency with which children are expected to use the tool – e.g., at least once daily? Will the pediatric oncology team members be informed of which children are participating in the study and when they are using PicPecc? Please clarify the timing of the parent and clinician interviews. Will these be occurring after each of the study periods? While the study protocol documents questions to be included in the workshops, it is unclear how clinicians caring for child participants will be included?
--	--

VERSION 2 – AUTHOR RESPONSE

Reviewer: 2

Comments

Regarding the secondary research question, when you speak of “alter stakeholders’ perspectives ...” are you proposing a specific direction in relation to these altered perspectives? Might you reframe this question slightly as to give attention to the desired direction?

Answers

We have clarified this on page 11, line 11-12.

(ii) Does the application of the PicPecc tool, alter stakeholders’ perspectives in a positive direction towards person-centred communication?

Comments

Please provide justification regarding the inclusion of children with relapsed disease.

Answers

Thank you for this comment. Upon reflection we see that if we include patients with relapsed disease we could introduce a confounding factor into the analysis that we would like to avoid. We have therefore removed this inclusion criteria and we will only include

patients diagnosed with ALL during the study period.

Comments

Please provide inclusion/exclusion criteria regarding the parents/guardians and healthcare providers.

Answers

We have clarified this on page 13, line 15-21.

The inclusion criteria for legal guardians will be that their child has undergone the high-dose methotrexate treatment and has used the PicPecc tool. In addition, legal guardians will need to be at the hospital during the treatment.

The inclusion criteria for healthcare providers will be that they are responsible for the children's care during the high-dose methotrexate treatment when the children use the PicPecc tool.

Comments

Will parents and children be enrolled as dyads? If a child is eligible yet a parent does not meet enrollment criteria, would the child be allowed to participate in the study. Which members of the pediatric oncology team are eligible to participate in the study and how will they be recruited?

Answers

We have clarified this on page 14 line 8-10, and page 18, line 3-6.

Healthcare providers at each of the units will be interviewed. The nurse and/or nurses that initiate and conclude the high-dose methotrexate treatment, will be invited to a semi-structured interview.

The child and the legal guardian are interviewed separately. The aim is to interview both the child and their legal guardian however, if either of them is unwilling or does not fit the criteria the other (child or legal guardian) will be invited to participate on their own.

Comments

Please address the following clarifying questions in relation to the informed consent/assent process:

- To clarify, will the children 15-17 years of age provide written consent along with written permission from their parents? In Sweden, is this regarded as written consent or written assent?

Answers

We have clarified this on page 14, line 19-21.

In Sweden, children between 15 and 18 years can provide written informed consent themselves if they are assessed to have the level of maturity and capacity to understand

the consequences of participation.

Comments

- Are the children less than 15 years of age required to provide any type of written assent or is verbal assent sufficient?

Answers

We have clarified this on page 14, line 18.

Upon consent from a legal guardian, child assent is obtained verbally and in writing if the child can read.

Comments

Thank you for providing information about the typical prevalence of children newly diagnosed with ALL in southern Sweden each year. Of the children with ALL, approximately how many are anticipated to be eligible to participate in the study, and what is the anticipated recruitment/enrollment rate?

Answers

We have clarified this on page 14, line 5-7.

At the three childhood cancer centres and at the five regional hospitals included in this study, approximately 25 of these children will fulfil the inclusion criteria for this study each year.